# Neuroprotective and Antiherpetic Properties of Polyphenolic Compounds from *Maackia amurensis* Heartwood

**DOI:** 10.3390/molecules28062593

**Published:** 2023-03-13

**Authors:** Darya V. Tarbeeva, Dmitry V. Berdyshev, Evgeny A. Pislyagin, Ekaterina S. Menchinskaya, Natalya Y. Kim, Anatoliy I. Kalinovskiy, Natalya V. Krylova, Olga V. Iunikhina, Elena V. Persiyanova, Mikhail Y. Shchelkanov, Valeria P. Grigorchuk, Dmitry L. Aminin, Sergey A. Fedoreyev

**Affiliations:** 1G.B. Elyakov Pacific Institute of Bioorganic Chemistry, Far-Eastern Branch of the Russian Academy of Science, 690022 Vladivostok, Russia; 2G.P. Somov Institute of Epidemiology and Microbiology, Rospotrebnadzor, 690087 Vladivostok, Russia; 3Federal Scientific Center of the East Asia Terrestrial Biodiversity (Institute of Biology and Soil Science), Far Eastern Branch, Russian Academy of Sciences, 690022 Vladivostok, Russia

**Keywords:** polyphenolic compounds, stilbenes, neuroprotection, antioxidant activity, antiherpetic activity

## Abstract

In this study, we isolated a new isoflavanostilbene maackiapicevestitol (**1**) as a mixture of two stable conformers **1a** and **1b** as well as five previously known dimeric and monomeric stilbens: piceatannol (**2**), maackin (**3**), scirpusin A (**4**), maackiasine (**5**), and maackolin (**6**) from *M. amurensis* heartwood, using column chromatography on polyamide, silicagel, and C-18. The structures of these compounds were elucidated by NMR, HR-MS, and CD techniques. Maksar^®^ obtained from *M. amurensis* heartwood and polyphenolics **1**–**6** possessed moderate anti-HSV-1 activity in cytopathic effect (CPE) inhibition and RT-PCR assays. A model of PQ-induced neurotoxicity was used to study the neuroprotective potential of polyphenolic compounds from *M. amurensis*. Maksar^®^ showed the highest neuroprotective activity and increased cell viability by 18% at a concentration of 10 μg/mL. Maackolin (**6**) also effectively increased the viability of PQ-treated Neuro-2a cells and the value of mitochondrial membrane potential at concentrations up to 10 μΜ. Maksar^®^ and compounds **1**–**6** possessed higher FRAP and DPPH-scavenging effects than quercetin. However, only compounds **1** and **4** at concentrations of 10 μM as well as Maksar^®^ (10 μg/mL) statistically significantly reduced the level of intracellular ROS in PQ-treated Neuro-2a cells.

## 1. Introduction

*Maackia amurensis* Rupr et Maxim. is an endemic woody plant of the Fabaceae family, widespread in the Primorsky and Khabarovsky regions of the Russian Federation. The heartwood of *M. amurensis* is included in the state register of medicinal products and its polyphenolic complex is used to produce the drug Maksar^®^. Maksar^®^ was developed at the G.B. Elyakov Pacific Institute of Bioorganic Chemistry (PIBOC FEB RAS) and registered in the Russian Federation as a hepatoprotective drug (registration number P N003294/01). The active compounds of this complex were isoflavones, pterocarpans, flavanones, isoflavans, isoflavanones, chalcones, lignans, and monomeric and dimeric stilbenes [1]. In contrast to *M. amurensis* heartwood, the main polyphenolic metabolites of *M. amurensis* root bark were shown to be glycosides of isoflavones and pterocarpans as well as prenylated flavanones [2,3]. 

In addition to possessing hepatoprotective properties, Maksar^®^ also reduced the total lipid content in blood [4,5]. It prevented the increase in the total serum lipid content and the development of hyperlipoproteinemia in experimental animals [6] and in clinical trials [7]. Maksar^®^ also possessed antiplatelet [8] and antitumor properties [9], as well as demonstrated antioxidant activity in vitro [10]. This drug enhanced the antioxidant defense system of the body and reduced the lipid peroxidation level [5,11].

The neuroprotective activity of the polyphenolic compounds from *M. amurensis* may be associated with their ability to inhibit monoamine oxidase (MAO), an enzyme that catalyzes the oxidative deamination of monoamines and exists in two isoforms: MAO-A and MAO-B. In the mammalian brain, MAO-B activity increases with aging. Hence, the inhibitors of this enzyme can be used to treat Parkinson′s disease [12,13]. 

The ability of 19 polyphenolic compounds from *M. amurensis* to inhibit MAO-B was studied, and the isoflavones calicosin and 8-*O*-methylretusin as well as the pterocarpan (—)-maakiain were shown to be effective MAO-B inhibitors [14,15]. The isoflavones genistein and formononetin significantly inhibited both MAO-A and MAO-B [15]. Maakiain was also shown to protect dopaminergic neurons from damage in several strains of transgenic worms [16]. 

The pathogenesis of neurodegenerative diseases, including Parkinson′s disease, involves the death of neurons from oxidative damage. An increase in the content of intracellular reactive oxygen species (ROS) causes DNA damage [17]. Many researchers also suggested a correlation between neurodegeneration and herpes infection [18,19]. Herpesvirus infections are also associated with the generation of oxidative stress in infected cells. It was reported that herpes simplex virus type 1 (HSV-1) caused an increase in the level of ROS and lipid peroxidation and also reduced the content of glutathione, the main tool of the body′s antioxidant defense [20,21]. It was shown that HSV-1 disrupted the functional capacity of mitochondria [22,23,24]. The impaired mitochondrial function, elevated ROS levels, and reduced mitochondrial membrane potential due to the herpes infection subsequently led to the development of neurodegenerative diseases [25,26,27]. 

The antiherpetic properties of the polyphenolics from *M. amurensis* have not been studied so far, except for resveratrol and piceatannol. Their antiherpetic activity was studied both in vitro and in vivo [28,29,30,31]. The aim of this study was to assess the antiherpetic activity of polyphenolic compounds, the constituents of Maksar^®^, as well as their ability to reduce the level of intracellular ROS and increase mitochondrial membrane potential using a model of Parkinson′s disease.

## 2. Results

Here, we isolated a new compound **1** as well as five previously known dimeric and monomeric stilbens: piceatannol (**2**), maackin (**3**), scirpusin A (**4**), maackiasine (**5**), and maackolin (**6**) from *M. amurensis* heartwood (Figure 1). Compounds **2**–**6** were identified by comparison of their HPLC-PDA-MS and NMR spectra with previously published data [32,33,34].

### 2.1. Structure Determination of Conformers ***1a*** and ***1b***

The mixture of compounds **1a** and **1b** was obtained as a white amorphous powder. We observed only one peak in the HPLC profile of the mixture of **1a** and **1b** (Appendix A). The molecular formulas of **1a** and **1b** were determined to be C_30_H_26_O_8_ based on the presence of [M-H]^−^ and [M+H]^+^ ions at *m*/*z* 513.1573 (calculated for [C_30_H_25_O_8_]^−^ 513.1555) and 515.1687 (calculated for [C_30_H_27_O_8_]^+^ 515.1700), respectively, in the HR-MS-ESI spectra of **1a** and **1b** (Appendix A). 

Sixteen carbon atoms belonged to the vestitol skeleton (rings A–C) and methoxy group of **1a**, whereas the other fourteen atoms formed a piceatannol moiety (rings D and E). The ^1^H NMR spectrum of **1a** showed the presence of an ABX spin system: the signals at *δ_H_* 6.64 (d, *J* = 8.4 Hz, 1H), 6.29 (dd, *J* = 8.4 Hz, 2.4 Hz. 1H), and 6.42 (d, *J* = 2.4 Hz, 1H) were attributed to protons H-5, H-6, and H-8 of ring A, respectively (Table 1). The signals of another ABX system at *δ_H_* 6.30 (d, *J* = 2.4 Hz, 1H), 6.27 (dd, *J* = 8.5 Hz, 2.4 Hz, 1H), and 7.25 (d, *J* = 8.5 Hz, 1H) in the ^1^H NMR spectrum of **1a** were assigned to H-3′, H-5′, and H-6′ of ring B in **1a**, respectively. A singlet at *δ_H_* 3.63 (s, 3H) in the ^1^H NMR spectrum of **1a** and a carbon signal at *δ_C_* 54.9 in the ^13^C NMR spectrum of **1a** were ascribed to OCH_3_-group at C-4′. We concluded that the methoxy group was located at C-4′, because we observed the correlation between the singlet signal of its protons at *δ_H_* 3.63 and the signal of C-4′ at *δ_C_* 159.8 in the HMBC spectrum of **1a** (Table 1). The proton signals at *δ_H_* 4.24 (dd, *J* = 10.3 Hz, 3.5 Hz 1H), 4.03 (t, *J* = 10.3 Hz, 1H), 4.16 (td, *J* = 11.6 Hz, 3.5 Hz, 1H), and 5.36 (d, *J* = 11.6 Hz, 1H) belonged to H-2a, H-2b, H-3, and H-4 of ring C in **1a**, respectively (Table 1).

Two doublets at *δ_H_* 6.82 (d, *J* = 16.0 Hz, 1H) and 6.57 (d, *J* = 16.0 Hz, 1H) were due to H-7″ and H-8″, which formed a double bond in the piceatannol moiety of **1a**. The value of the coupling constant (*J* = 16.0 Hz) confirmed *trans* configuration of the double bond. Two doublets in the ^1^H NMR spectrum of **1a** at *δ_H_* 6.26 (d, *J* = 2.5 Hz, 1H) and 6.51 (d, *J* = 2.5 Hz, 1H) were assigned to H-3″ and H-5″ of ring D in **1a**, respectively. The ABX-system of ring E in **1a** was formed by H-10″, H-13″, and H-14″, which gave signals at *δ_H_* 6.80 (d, *J* = 2.4 Hz, 1H), 6.73 (d, *J* = 8.1 Hz, 1H), and 6.67 (dd, *J* = 8.1 Hz, 2.4 Hz 1H) in the ^1^H NMR spectrum of **1a**, respectively.

The carbon atom C-4 (*δ_C_* 35.8) of the vestitol moiety in **1a** was linked to C-1″ (*δ_C_* 119.5) of the piceatannol moiety, because in the HMBC spectrum of **1a** we observed cross-peaks between the proton signal at *δ_H_* 5.36 (d, *J* = 11.6 Hz, 1H) of H-4 and the carbon signals of C-1″, C-2″, and C-6″at *δ_C_* 119.5, 157.4, and 139.6, respectively. 

The chemical shift values in the ^1^H and ^13^C spectra of compound **1a** did not differ considerably from those of **1b** except for C-4. The vestitol (rings A–C) skeleton, including methoxy group, and piceatannol (rings D and E) moiety in **1b** were formed by other sixteen and fourteen carbon atoms, respectively. The ^1^H NMR spectrum of **1b** revealed the presence of an ABX spin system: the signals at *δ_H_* 6.50 (d, *J* = 8.8 Hz, 1H), 6.20 (dd, *J* = 8.8 Hz, 2.4 Hz. 1H), and 6.27 (d, *J* = 2.4 Hz, 1H) were due to protons H-5, H-6, and H-8 of ring A, respectively (Table 2). The protons of another ABX system gave signals at *δ_H_* 6.40 (d, *J* = 2.4 Hz, H), 6.22 (dd, *J* = 8.5 Hz, 2.4 Hz, 1H), and 6.99 (d, *J* = 8.5 Hz, 1H) in the ^1^H NMR spectrum of **1b** (H-3′, H-5′, and H-6′ of ring B in **1b**, respectively). A singlet at *δ_H_* 3.61 (s, 3H) in the ^1^H NMR spectrum of **1b** and a carbon signal at *δ_C_* 54.9 in the ^13^C NMR spectrum of **1b** were ascribed to the OCH_3_-group at C-4′. The correlation between protons at *δ_H_* 3.61 and the carbon signal at *δ_C_* 159.9 in the HMBC spectrum of **1b** confirmed that the methoxy group was located at C-4′ (Table 2). The signals at *δ_H_* 4.20 (dd, *J* = 10.3 Hz, 3.2 Hz 1H), 4.44 (t, *J* = 10.3 Hz, 1H), 4.36 (td, *J* = 10.9 Hz, 3.2 Hz, 1H), and 4.95 (d, *J* = 10.9 Hz, 1H) were assigned to H-2a, H-2b, H-3, and H-4 of ring C in **1b**, respectively (Table 2).

Two doublets at *δ_H_* 7.25 (d, *J* = 15.9 Hz, 1H) and 6.62 (d, *J* = 15.9 Hz, 1H) were due to H-7″ and H-8″, which formed a double bond in the piceatannol moiety of **1b**. The value of the coupling constant (*J* = 15.9 Hz) also confirmed *trans* configuration of the double bond in the piceatannol moiety of **1b**. Two doublets in the ^1^H NMR spectrum of **1b** at *δ_H_* 6.20 (d, *J* = 2.4 Hz, 1H) and 6.44 (d, *J* = 2.4 Hz, 1H) were assigned to H-3″ and H-5″ of ring D in **1b**, respectively. The ABX-system of ring E in **1b** was formed by H-10″, H-13″, and H-14″, which gave signals at *δ_H_* 7.09 (d, *J* = 1.9 Hz, 1H), 6.78 (d, *J* = 8.5 Hz, 1H), and 6.84 (dd, *J* = 8.5 Hz, 1.9 Hz 1H) in the ^1^H NMR spectrum of **1b**, respectively. C-4 (*δ_C_* 39.7) of the vestitol moiety of **1b** was also linked to C-1″ (*δ_C_* 119.4) of the piceatannol moiety, because the cross-peaks between the proton signal at *δ_H_* 4.95 (d, *J* = 10.9 Hz, 1H) of H-4 and the carbon signals of C-1″, C-2″, and C-6″ at *δ_C_* 119.4, 157.1, and 141.7, respectively, were observed in the HMBC spectrum of **1b**.

All signals in the ^1^H and ^13^C NMR spectra of compounds **1a** and **1b** were completely assigned on the basis of COSY, HMBC, and ROESY spectral data. The NMR spectra for **1** can be found in Appendix A. Thus, compounds **1a** and **1b** were determined to be dimeric compounds composed of vestitol moiety and piceatannol moiety. 

In order to determine the absolute configuration of compound **1**, we used the approach based on the combination of theoretical and experimental UV, ECD, and NMR spectroscopy methods. 

The experimental NMR spectra (*δ_H_*, *J_H_*_–*H*_, and ROESY data) definitely gave evidence that the sample of **1** under study was a mixture of compounds, which might be two different stereoisomers of **1** or two different conformers of one stereoisomer. Thus, we had to distinguish between these two possibilities.

We performed the extended quantum-chemical investigation of the conformational mobility of 3*R*,4*S* and 3*S*,4*S* stereoisomers of **1** using density functional theory (DFT) with the B3LYP exchange-correlation functional set, polarizable continuum model (PCM), and 6-311G(d) basis set implemented in the Gaussian 16 package of programs [35]. The details of the calculations are described in Appendix A. Many large amplitude motions (LAM) may proceed in each stereoisomer: internal rotations of different hydroxyl groups, the internal rotation of the 4-vinylbenzene-1,2-diol fragment of the piceatannol substituent at C-4, the inversion of ring C, and the internal rotations of substituents at C-3 and C-4. Due to these intramolecular motions, a number of rotameric forms for each conformation of ring C may occur for both 3*R*,4*S* and 3*S*,4*S* configurations. Some of them are favorable for the creation of intramolecular O-H…O hydrogen bonds (Appendix A), which stabilize these rotameric forms for about several kcal/mol relative to other rotameric forms. 

The values of vicinal spin–spin coupling constants *J_H4__−__H3_*, *J_H2a__−__H3_*, and *J_H2b__−__H3_* strongly depend on the values of dihedral angles θ_4_ ≡ ∠ H-4–C-4–C-3–H-3, θ_2a_ ≡ ∠ H-2a–C-2–C-3–H-3, and θ_2b_ ≡ ∠ H-2b–C-2–C-3–H3, respectively. According to the Karplus equation, *J_Hi__−__H3_* ≈ 10–11 Hz when θ*_i_* ≈ 0 ± 20° or 180 ± 20°. The values of *J_H4–H3_* in the experimental ^1^H NMR spectra of **1a** and **1b** were 11.6 Hz and 10.9 Hz, respectively (Table 1 and Table 2).

According to the results of the conformational analysis, the values of dihedral angle θ_4_ for stable conformations of 3*S*,4*S* stereoisomer of **1** varied in the diapason of |θ_4_| ≥ 35°. The most abundant conformations of 3*S*,4*S* stereoisomer of **1** with total statistical weight of more than 93% (Appendix A) arose when θ_5_ ≡ ∠ C-2″–C-1″–C-4–H-4 ≈ 180 ± 30°. The most stable ones with total amount ≈ 81% arose when ring C had conformation with axial orientation (Ax) of the atom H-3 (θ_4_ ≈ −43°). In the most stable conformations, the atom H-7″ stayed in the proximity to the atom H-4 (the distance (H-4…H-7″) ≈ 2.1 Å) and far from the atom H-3 (the distance (H-3…H-7″) ≈ 3.4 Å). The second most stable conformation arose when ring C had conformation with equatorial (Eq) orientation of the atom H-3 (θ_4_ ≈ +36°) and the atom H-7″ stayed far from both the H-3 and the H-4 atoms (the distance (H4…H-7″) ≈ 3.8Å and the distance (H3…H-7″) ≈ 4.1Å). Based on these geometries, we were able to suppose that 3*S*,4*S*- stereoisomer of **1** poorly matched the experimental values of *J_H4__−__H3_* constants and the ROESY data. To prove this, we calculated these constants quantum-chemically using the B3LYP/6-311G(d)_GIAO_PCM//B3LYP/6-311G(d)_PCM level of theory. For 3*S*,4*S*-**1**_Ax, the obtained values were: *J_H4–H3_* = +6.5 Hz, *J_H3–H2a_* = +3.1 Hz, and *J_H3–H2b_* = +9.4 Hz. For 3*S*,4*S***-1**_Eq, the conformation theoretical values were: *J_H-4–H-3_* = +8.1 Hz, *J_H-3–H-2a_* = +3.7 Hz, and *J_H3–H2b_* = +1.1 Hz. The statistically averaged values were: *J_H4–H3_* = +6.2 Hz, *J_H3–H2a_* = +2.9 Hz, and *J_H3–H2b_* = +7.7 Hz. Thus, the theoretical NMR data for the 3*S*,4*S*- stereoisomer of **1** (and, hence, for the 3*R*,4*R*-stereoisomer as well), did not match the experimental NMR data.

Two main intramolecular rearrangements were considered when analyzing the NMR, UV, and ECD spectra of 3*S*,4*R* stereoisomer of **1** (Figure 2).

The big size of the 4-vinylbenzene-1,2-diol fragment at C6″ resulted in high steric hindrances, which inhibited the internal rotation of piceatannol fragment around C4−C1″ bond. The scans of the potential energy surface along θ(6″) ≡ ∠ 6″ − 1″ − C-4 − H-4 are given in Appendix A. Our calculations showed that 3*R,*4*S*-**1a** ↔3*R,*4*S*-**1b** rearrangement could proceed only when ring C had equatorial (“Eq”) conformation. Additionally, even in this case, the transfer was required to overcome the wide and high potential energy barrier: ΔV^#^*_1a__↔__1b_* ≥ 20.0 kcal/mol. The rate constant calculated for this ΔV^#^*_1a__↔__1b_* value was *k_1a↔__1b_* = ≤ 10^−1^ c^−1^.

The inverse value of *k_1a__↔__1b_* was a lifetime τ of conformations **1a** and **1b**. The lifetimes τ ≈ 10^1^ ÷ 10^2^ c were about 3 ÷ 8 orders of magnitude larger than the characteristic time in NMR experiments was. Thus, the conditions of the very slow exchange between two stable states were fulfilled and we were able to detect experimentally two different rotameric forms (**1a** and **1b**) of the 3*R*,4*S* stereoisomer of **1** by NMR technique. 

In contrast to the **1a**↔**1b** rotamerism, the inversion of ring C proceeded with overcoming of relatively low potential energy barrier: ΔV^#^_inv_ ≥ 4 kcal/mol (Appendix A). The rate constant for this process was *k_inv_*~10^9^ ÷ 10^10^ c^−1^ and τ_Ax/Eq_ ≈ 10^−10^ ÷ 10^−9^ c. These lifetimes corresponded to the conditions of quick exchange and, hence, conformations, differing in the conformation of ring C, could not be distinguished by NMR method.

The performed conformational analysis (accounting for rotamerism of substituents at C-3 and C-4, rotamerism of OH groups, the inversion of ring C and the formation of O−H…O intramolecular hydrogen bonds) allowed us to select conformations with Gibbs free energies in the region ΔG_im_ ≤ 5 kcal/mol. The structures of these conformations and their statistical weights are presented in Appendix A. According to the performed analysis, the 3*R,*4*S*-**1a** and 3*R,*4*S*-**1b** rotameric forms should be treated as two different compounds. According to the calculated Gibbs free energies, compound 3*R,*4*S*-**1a** existed predominantly in “Ax” conformation, whereas compound 3*R,*4*S*-**1b** existed as a mixture of “Ax” and “Eq” conformations in the ratio g(Ax): g(Eq) ≈ 1.63. 

The “Ax” conformation of 3*R,*4*S*-**1a** was characterized by dihedral angles θ_4_ ≈ 180°, θ_2a_ ≈ +175°, and θ_2b_ ≈ −64°. This geometry coincided well with the geometry, which might be expected according to relative values of experimental *J_H-3-H-4_*, *J_H-3-H-2a_*, and *J_H-3-H-2b_* constants. In this conformation, the atom H-3 stayed in the proximity to atoms H-7″ (the distance (H3…H-7″) ≈ 2.4 Å), whereas H-4 stayed in the proximity to atoms H-6′ (the distance (H4…H-6′) ≈ 2.1 Å). These data also were in accordance with the ROESY experiments (Table 1 and Table 2).

The “Ax” conformation of 3*R,*4*S*-**1b** was characterized by dihedral angles θ_4_ ≈ −174°, θ_2a_ ≈ +173°, and θ_2b_ ≈ −65°. This geometry also coincided well with the relative values of experimental *J_H3-H4_*, *J_H3-H2a_*, and *J_H3-H2b_* constants (Table 1 and Table 2). In this conformation, the atom H-4 stayed in the proximity to atoms H-7″ (the distance (H4…H-7″) ≈ 2.0 Å) and the atom H-3 stayed in the proximity to atoms H-2a (the distance (H3…H-2a) ≈ 2.4 Å). These data also were in accordance with the ROESY data (Table 1 and Table 2). 

Then, we calculated theoretical values of vicinal spin–spin coupling constants for both stable rotameric forms of 3*R,*4*S* stereoisomer of **1**:

3*R,*4*S*-**1a**_Ax: *J_H-4__−__H-3_* = +10.4 Hz, *J_H3__−__H2a_* = +9.4 Hz, *J_H3__−__H2b_* = +3.2 Hz;

3*R,*4*S*-**1b**_Ax: *J_H-4__−__H-3_* = +9.7 Hz, *J_H3__−__H2a_* = +8.9 Hz, *J_H3__−__H2b_* = +2.9 Hz; 

3*R,*4*S*-**1b**_Eq: *J_H-4__−__H-3_* = +5.7 Hz, *J_H3__−__H2a_* = +2.1 Hz, *J_H3__−__H2b_* = +2.7 Hz;

The statistically averaged values for 3*R,*4*S* stereoisomer (**1b**) were: *J_H-4__−__H-3_* = +7.8 Hz, *J_H3__−__H2a_*= +6.0 Hz, and *J_H3__−__H2b_* = +2.7 Hz.

Thus, the theoretical NMR data calculated for the 3*R,*4*S* stereoisomer of **1a** corresponded well for one set of signals in the experimental NMR ^1^H spectrum (Table 1). On the contrary, for **1b**, the theoretical values of vicinal spin–spin coupling constants were smaller than the experimental values. The relative values of *J_H-4__−__H-3_*, *J_H3__−__H2a_,* and *J_H3__−__H2b_* constants were reproduced correctly (Table 2).

The linear regression analysis showed that calculated *J* values were systematically underestimated:

*J_calc_*(3*R*,4*S*-**1a**) = 0.017 + 0.879 · *J_exp_*;

*J_calc_*(3*R*,4*S*-**1b**_Ax) = -0.013 + 0.839 · *J_exp_*. 

When scaling factor η = 1/0.839 = 1.192 was used, the recalculated mean values of constants for 3*R*,4*S*-**1b** became: *J_H-4–H-3_* = +9.3 Hz, *J_H3–H2a_* = +7.2 Hz, *J_H3–H2b_* = +3.2 Hz. These values were still lower than the experimental values. Thus, we were able to suppose that the theory we used overestimated to some extent the amount of 3*R*,4*S*-**1b**_Eq conformations.

According to NMR data, the concentrations of **1a** and **1b** in the sample related as η ≈ 1: 0.85. Figure 3 shows the averaged ECD spectrum, calculated for the 3*R*,4*S* stereoisomer of **1** as a superposition of spectra, obtained for 3*R*,4*S*-**1a** and 3*R*,4*S*-**1b**. The variations of Δε (3*R*,4*S*-**1**) contour dependent on relative amounts of “Ax” and “Eq” conformations are presented in Appendix A. A good correspondence between theoretical and experimental spectra was obtained for the 3*R*,4*S*-**1** stereoisomer, but a poor one for the 3*S*,4*R*-**1** stereoisomer. These data confirmed that the absolute configuration of **1** was 3*R*,4*S*.

Based on the results of the performed quantum-chemical calculations and NMR data, we found that compound **1** was a mixture of two stable conformers **1a** and **1b** with 3*R*,4*S* absolute configuration (Figure 2). Hence, compound **1** was named maackiapicevestitol and its structure was determined to be 4-(*E*)-3,5-dihydroxy-2-[(*3R*,4*S*)-7-hydroxy-3-(2-hydroxy-4-methoxyphenyl)-chroman-4-yl)-styryl]-benzene-1,2-diol.

The absolute configurations of compounds **3**, **4**, and **6** were determined by comparison of the experimental and theoretically calculated ECD spectra. The details of the procedure were the same as for **1**. First of all, we used the B3LYP/6-311G(d)_PCM method to investigate the relative thermodynamic stability of different conformations for all compounds. The most stable conformations were selected for further calculation of the UV and ECD spectra. The TDDFT approach along with the cam-B3LYP density functional were used for calculation of excitation energies and the rotatory and oscillatory strengths for a number of vertical electronic transitions (n = 50 for each conformation). The comparison of the calculated UV spectra with the experimental ones was used to obtain the UV shifts and the bandwidths that gave good coincidence between theoretical and experimental UV spectra. 

The analysis of the theoretically obtained UV spectra (Appendix A) showed that the TDDFT_cam-B3LYP/6-311G(d)_PCM//B3LYP/6-311G(d)_PCM method allowed us to reproduce well experimental UV spectra in the short-wave region λ ≤ 260 nm. These data forced us to choose the λ ≤ 260 nm region as a reference region for determination of the UV shifts and bandwidths for all compounds. 

The spectra calculated for compound **3** are presented in Appendix A. 

The UV shift Δλ = +12 nm and the bandwidth ζ = 0.38 eV were used for simulations of the UV and ECD spectra. The comparison of experimental and theoretical ECD spectra showed that the correct sign of the band in λ ≤ 260 nm region reproduced the ECD spectrum, calculated for 2*S*,3*S* stereoisomer of **3**.

Appendix A shows the theoretical and experimental UV and ECD spectra of **6** (the UV shift Δλ = +10 nm and the bandwidth ζ = 0.32 eV were used). Good correspondence between calculated and experimental ECD spectra occurring for 3*S*,3a*S*,8*R*,8a*S* configuration of **6** was observed. 

The determination of the configurations of asymmetric centers for compounds **4** and **5** is in progress.

### 2.2. Antiradical Activity and Ferric Reducing Power (FRAP) of Polyphenolic Compounds from M. amurensis Heartwood

Compounds **1**–**6** exhibited considerably higher DPPH-scavenging effect and FRAP compared to those of the reference compounds quercetin and ascorbic acid (Table 3). In the DPPH assay, the IC_50_ values for **1**–**6** were in the range from 2.0 to 4.3 µM compared to 9.3 µM for quercetin. Maackolin (**6**) showed the highest antiradical activity (IC_50_ 2.0 µM) among the tested compounds, as well as significant FRAP (23.10 C_Fe2+_(µM)/C_maackolin_). Maackiasine (**5**) exhibited quite significant DPPH-scavenging effect, but its FRAP was the lowest among the tested stilbenes. The DPPH-scavenging effect and FRAP of Maksar^®^ were lower than those of individual polyphenolics **1**–**6** (Table 3).

### 2.3. Cytotoxic Activity of Stilbenes from M. amurensis against Neuro-2a Cells

The evaluation of cytotoxic activity of stilbenes **1**, **2**, **4**–**6** against Neuro-2a cells showed that they did not affect cell viability at concentrations up to 100 μM. It was also shown that only maackin (**3**) at the maximum studied concentration of 100 μM reduced cell viability with EC_50_ value of 87.7 µM (data not shown).

### 2.4. Effect of Polyphenolic Compounds from M. amurensis Heartwood on the Viability and ROS Level in PQ-Treated Neuro-2a

In this study, we performed the MTT assay to assess the percentage of living cells after treatment with paraquat (PQ). Maksar^®^ showed the highest neuroprotective activity and increased cell viability by 18% at a concentration of 10 μg/mL compared to PQ-treated Neuro-2a cells (Figure 4a,b). Maackolin (**6**), at a concentration of 10 μM, also effectively increased the viability of PQ-treated cells (by 16%). Piceatannol (**2**) and maackiasine (**5**) did not show any effect in this test. Maackiapicevestitol (**1**) and scirpusin A (**4**), at a concentration of 10 μM (5.1 and 4.9 μkg/mL, respectively), as well as Maksar^®^ (10 μg/mL) reduced the level of intracellular ROS in PQ-treated Neuro-2a cells by 17%, 7%, and 12%, respectively (Figure 4c). Maackin (**3**) at the minimum studied concentration of 0.1 μM inhibited the level of ROS by 17%. Maackolin (**6**) did not decrease the ROS level in PQ-treated Neuro-2a cells (Figure 4c).

### 2.5. Effect of Polyphenolic Compounds from M. amurensis Heartwood on Mitochondrial Membrane Potential in PQ-Treated Neuro-2a

We studied the effect of stilbenes from *M. amurensis* as well as Maksar^®^ on PQ-induced mitochondrial dysfunction in Neuro-2a cells. The tetramethylrhodamine methyl (TMRM) fluorescence decreased by 16% after a 1 h exposure of Neuro-2a cells with PQ, which indicated that PQ caused depolarization of the mitochondrial membrane (Figure 4d). All tested compounds were able to prevent depolarization and restored mitochondrial membrane potential to almost baseline values. We observed a significant dose-dependent effect of increasing mitochondrial potential when Neuro-2a cells were incubated with scirpusin A (**4**) compared to PQ-treated cells. The maximum effect was observed when Neuro-2a cells were incubated with **4** at a concentration of 10 μM (26%). Maackolin (**6**) at a concentration of 10 μM also increased the value of mitochondrial membrane potential compared to PQ-treated cells by 9%. When maackin (**3**) was added to PQ-treated Neuro-2a cells, an inverse dose-dependent effect was observed. The most effective concentration was 0.1 μM (the value of mitochondrial membrane potential increased by 16% compared with PQ-treated cells) (Figure 4d).

### 2.6. Cytotoxic Activity of Stilbenes from M. amurensis against Vero Cells

The study of cytotoxicity of polyphenolic complex from *M. amurensis* heartwood against Vero cells was carried out using the MTT assay. Maksar^®^ showed the lowest cytotoxicity (CC_50_ > 1200 μg/mL) (Table 4). Piceatannol (**2**) also demonstrated low cytotoxicity (CC_50_ > 1020 µM). The cytotoxicity of polyphenolic compounds **3**–**6** ranged from 300 to 560 μM. To study the antiviral effect of stilbenes **1**–**6**, we applied the compounds at concentrations that were below the CC_50_ values. 

### 2.7. Anti-HSV-1 Activity of Polyphenolic Compounds from M. amurensis (CPE assay)

The antiviral activity of stilbenes from *M. amurensis* against HSV-1 was assessed using the cytopathic effect inhibition (CPE) assay. To study the inhibitory effect of polyphenolic compounds on the early stage of viral infection, these were added to Vero cells simultaneously with the virus. Maksar^®^ inhibited virus replication more effectively than stilbenes **1**–**6** (IC_50_ and SI values were 13.9 µg/mL and 87, respectively; Table 4). Polyphenolic compounds **1**–**4** also showed significant antiviral activity (IC_50_ values ranged from 27 to 90 µM). Maackiasine (**5**) and maackolin (**6**) showed moderate activity with SI values of 6.6 and 6.4 and IC_50_ values of 45.0 and 88.9, respectively (Table 4).

### 2.8. Anti-HSV-1 Activity of Polyphenolic Compounds from M. amurensis (RT-PCR)

The anti-HSV-1 activity of the polyphenolic compounds from *M. amurensis* was also studied using the real-time PCR (RT-PCR) technique (Figure 5 and Appendix A). Vero cells were simultaneously infected with 100 TCD_50_/_mL_ HSV-1 and treated with polyphenolic compounds **1**–**6** at concentrations of 50 and 5 μg/mL. After 48 h of cell incubation, viral DNA was extracted from the supernatants, and the relative level of HSV-1 DNA was determined using RT-PCR. The effect of stilbenes on the relative level of viral DNA was assessed by 2^−ΔCt^ method and reported as fold reduction relative to virus control, to which the value of 1.0 was assigned.

We showed that polyphenolic compounds from *M. amurensis* significantly inhibited the replication of HSV-1 at a concentration of 50 μg/mL, when Vero cells were simultaneously treated with these compounds and infected with HSV-1 (Figure 5 and Appendix A). At this concentration, compounds **1**–**4** and Maksar^®^ caused a 4.2 log10 reduction in viral DNA compared to virus control (*p* < 0.001). However, at a concentration of 5 µg/mL, only compounds **1**, **4**, and Maxar^®^ reduced the relative level of HSV-1 DNA by 1.3 log10 relative to virus control on average (*p* < 0.05). At the same time, the reference medicine acyclovir, at a concentration of 5 µg/mL, caused a 4 log10 reduction in the viral DNA (data not shown). 

The obtained results showed that the tested polyphenolic compounds inhibited HSV-1 infection in a dose-dependent manner in both the CPE inhibition and RT-PCR assays. We found that compounds **1**–**4** and Maksar^®^ were the most effective in reducing viral replication when added simultaneously with the initiation of viral infection.

## 3. Discussion

In this study, we investigated antioxidant, neuroprotective, and antiherpetic activities of polyphenolic compounds that were constituents of Maksar^®^ obtained from *M. amurensis* heartwood. The DPPH-scavenging effect and FRAP of polyphenolic compounds **1**–**6** were significantly higher compared to the reference compounds quercetin and ascorbic acid (Table 3).

In order to evaluate the neuroprotective properties of polyphenolic compounds from *M. amurensis* heartwood, including their ability to reduce intracellular ROS level and increase mitochondrial membrane potential, we used a model of PQ-induced neurotoxicity in Neuro-2a cells. Maksar^®^ showed the highest neuroprotective activity and increased cell viability by 18% at a concentration of 10 μg/mL compared to PQ-treated cells. Stilbenolignan maackolin (**6**), at a concentration of 10 μM, also effectively increased the viability of PQ-treated Neuro-2a cells (by 16%) at a concentration up to 10 μM, which may be due to the ability of this compound to increase the value of mitochondrial membrane potential. Although compounds **1**–**6** possessed high DPPH-scavenging effect and FRAP values, only compounds **1** and **4** at a concentration of 10 μM as well as Maksar^®^ (10 μg/mL) statistically significantly reduced the level of intracellular ROS in PQ-treated Neuro-2a cells. Dimeric stilbene scirpusin A (**4**) effectively increased mitochondrial membrane potential. Maackin (**3**) at the minimum studied concentration of 0.1 μM significantly inhibited the level of ROS and increased mitochondrial membrane potential. The inverse dose-dependent effect was mainly due to the cytotoxic activity of **3** against Neuro-2a cells. Monomeric stilbene piceatannol (**2**) and isoflavonostilbene maackiasine (**5**) did not significantly increase the viability of Neuro-2a cells.

In addition to the neuroprotective properties, polyphenolic compounds **1**–**6** showed moderate antiherpetic activity. We found that these compounds affected the early stage of the HSV-1 life cycle. Some studies also reported multiple mechanisms of anti-HSV-1 action of stilbenes isolated from various plants, including inhibition of virus adsorption and entry, reduction in gene expression, inhibiting the late viral protein synthesis, and stimulation of ROS production [29,36,37,38]. However, the most probable mechanism of antiherpetic activity of Maksar^®^ and its components **1** and **4** (Table 4, Figure 2) may be due to their ability to reduce ROS level, induced by HSV-1 in cells [39]. Hence, further study of the mechanisms of HSV-1 inhibition by stilbenes from *M. amurensis* is necessary.

## 4. Materials and Methods

### 4.1. Plant Material

*M. amurensis* was collected in September 2021 by academician P.G. Gorovoy (Andreevka village, Khasansky District) of the Primorsky region (Russian Far East). Voucher specimen (No. 103539) was deposited into the herbarium of the Laboratory of Chemotaxonomy (G.B. Elyakov Pacific Institute of Bioorganic Chemistry, FEB RAS).

### 4.2. Extraction and Isolation

We extracted the heartwood of *M. amurensis* (350 g) twice under reflux with a CHCl_3_–EtOH solution system (3:1, *v*/*v*) for 3 h at 60 °C. The obtained extract (11 g) was subjected to a polyamide column (100 g, 50–160 µm, Sigma-Aldrich, St. Louis, MI, USA). The column was eluted with a hexane–CHCl_3_ solution system with gradually increasing CHCl_3_ amounts (hexane/CHCl_3_, 1:0, 10:0, 5:1, 2:1, 1:1, 1:2) to obtain fractions 1–6 and then, with a CHCl_3_–EtOH solution system with gradually increasing EtOH amounts (CHCl_3_/EtOH, 1:0, 100:1, 50:1, 40:1, *v*/*v*) to obtain fractions 7–16.

We subsequently purified the fractions containing stilbenes according to the HPLC data. Fraction 12 (CHCl_3_/EtOH, 1:1, 1.16 g) (Appendix A) was subsequently subjected to a silica gel column (40–63 µm) and eluted with a hexane–CHCl_3_ solution system with gradually increasing CHCl_3_ amounts (hexane/CHCl_3_, 20:0, 10:1, 8:1, 5:1, *v*/*v*) twice to obtain the mixture of compounds **5** and **6** (27.6 mg). This mixture was then applied to a C-18 column to obtain individual compounds **5** (11.5 mg), and **6** (13.6 mg). 

Fraction 14 (EtOH, 940 mg) (Appendix A) was chromatographed twice over a silica gel column (40–63 µm, Sigma-Aldrich, St. Louis, MI, USA). The column was eluted with a CHCl_3_–EtOH solution system with gradually increasing EtOH amounts (CHCl_3_/EtOH, 1:0, 200:1, 100:1, 50:1, 40:1, *v*/*v*) to yield compounds **1** (9.1 mg), **2** (28.7 mg), and the mixture of compounds **3** and **4** (32.5 mg). Compounds **1** and **2** were subsequently purified using a C-18 column. The mixture of compounds **3** and **4** was then applied to a C-18 column to obtain individual compounds **3** (15.1 mg) and **4** (14.4 mg).

*Maackiapicevestitol* (**1**): white, amorphous powder; UV (MeOH) λ_max_ 202, 284, 332 nm; ECD (2.55 × 10^–4^ M, MeOH) λ_max_ (Δε) 202 (+2.01), 215 (−4.26), 236 (−1.99), 258 (+0.13), 280 (−0.48), 300 (+0.89), 322 (+0.78); ^1^H and ^13^C NMR data, see Table 1 and Table 2; HR-ESI-MS *m*/*z* 513.1573 [M-H]^−^ (calculated for [C_30_H_25_O_8_]^−^ 513.1555), *m*/*z* 515.1687 [M+H]^+^ (calculated for [C_30_H_27_O_8_]^+^ 515.1700).

### 4.3. General Experimental Procedures

We measured the UV spectra using a UV-1601 PC spectrophotometer (Shimadzu, Kyoto, Japan). The CD spectra were obtained on a Chirascan-plus Quick Start CD Spectrometer (Applied Photophysics Limited, Leatherhead, UK) (acetonitrile, 20 °C). We recorded the ^1^H, ^13^C, and two-dimensional NMR spectra in acetone-*d*_6_ using a Bruker AVANCE III DRX-700 NMR instrument (Bruker, Karlsruhe, Germany). 

### 4.4. HPLC Analysis

We used an Agilent Technologies 1260 Infinity II HPLC system (Agilent Technologies, Waldbronn, Germany) equipped with a VWD detector (λ = 280 nm) to perform the HPLC analysis of extracts and fractions. The flow rate was 0.8 mL/min using a Supelco Analytical HS-C18 (Supelco Analytical, Bellefonte, PA, USA). The column (3 μm, 4.6, 75 mm) was thermostated at 30 °C. The mobile phase consisted of 1% aqueous acetic acid (A) and acetonitrile containing 1% acetic acid (B). The following gradient steps were programmed: 0–2 min—10% B, 2–4 min—10–20% B, 4–21 min—20–30% B, 21–26 min—30–40% B, 26–31 min—40–50% B, 31–34 min—50–90% B, and 34–36 min—90–50% B. The data were analyzed using OpenLab CDS software v. 2.4 (Agilent Technologies, Waldbronn, Germany).

### 4.5. HR-ESI-MS Analysis

We recorded HR-ESI-MS spectra on a Shimadzu hybrid ion trap–time of flight mass spectrometer (Shimadzu, Kyoto, Japan). The electrospray ionization (ESI) source potential was 3.8 and 4.5 kV for the negative and positive ion mode, respectively; the drying gas (N_2_) pressure was 200 kPa; the nebulizer gas (N_2_) flow was 1.5 L/min; the temperature for the curved desolvation line (CDL) and heat block was 200 °C; the detector voltage was 1.5 kV, and the detection range was 100–900 *m*/*z*. The mass accuracy was below 4 ppm. Shimadzu LCMS Solution software (v.3.60.361, Shimadzu, Kyoto, Japan) was used to acquire and process the data.

### 4.6. Antiradical Activity

The DPPH (2,2-diphenyl-1-picrylhydrazyl) scavenging effect of polyphenolic compounds **1**-**6** from *M. amurensis* heartwood was evaluated as described previously in [40]. Stilbenes were added to DPPH solution in MeOH (10^−4^ M) at a concentration range from 1 to 34 µM. The mixture was kept in the dark at room temperature for 20 min. Then, we measured the absorbance at 517 nm using a Shimadzu UV 1240 spectrophotometer (Shimadzu, Kyoto, Japan). The DPPH radical-scavenging effect (%) of the stilbenes was calculated using Equation (1): (1)$$\mathrm{DPPH}\;\mathrm{scavenging}\;\mathrm{effect},\;\%=\frac{A_{0}-A_{x}}{A_{0}}\times 100,$$where: *A_0_* is the absorbance of DPPH solution without polyphenolic compounds (blank sample);*A_x_* is the absorbance of DPPH solution in the presence of different concentrations of polyphenolic compounds.

Quercetin and ascorbic acid were used as reference compounds. All experiments were performed in triplicate. The half maximal inhibitory concentration (IC_50_) for polyphenolic compounds was calculated by plotting the DPPH scavenging effect (%) against the concentrations of polyphenolic compounds. IC_50_ values are given as the mean ± SEM.

### 4.7. Ferric Reducing Antioxidant Power (FRAP) Assay

The FRAP assay was carried out as described in [41]. We prepared FRAP reagent by mixing 2.5 mL of TPTZ (2,4,6-tris(2-pyridyl)-s-triazine) solution (10 mM) in 40 mM HCl and 25 mL of FeCl_3_ solution (20 mM) in acetate buffer solution (300 mM, pH 3.6). Then, we added stilbenes **1**–**6** to 3 mL of FRAP reagent at concentrations from 1 to 34 µM. The mixture was kept in the dark at room temperature for 4 min. Then, we measured the absorbance at 595 nm using a Shimadzu UV 1240 spectrophotometer. Equation (2) was used to calculate the FRAP values of stilbenes **1**–**6**:(2)$$\mathrm{FRAP}=\frac{C_{Fe}}{C_{x}},$$
where:*C_Fe_* is the concentration of Fe^2+^ (µM) formed in the reaction;*C_x_* is the concentration of polyphenolic compounds in the reacting mixture.

The concentration of Fe^2+^ (µM) formed in the reaction was determined using the calibration curve obtained for different concentrations of FeSO_4_·7H_2_O.

### 4.8. Neuro-2a Cell Line and Culture Conditions

We purchased the murine neuroblastoma cell line Neuro-2a (CCL-131) from the American Type Culture Collection (ATCC^®^) (Manassas, VA, USA). We cultured Neuro-2a cells in Dulbecco′s Modified Eagle Medium (Biolot, St. Petersburg, Russia), which contained 10% fetal bovine serum (Biolot, St. Petersburg, Russia) and 1% penicillin/streptomycin (Biolot, St. Petersburg, Russia). We incubated the cells at 37 °C in a humidified atmosphere containing 5% (*v*/*v*) CO_2_.

### 4.9. The Viability of Neuro-2a Cells

We prepared the stock solutions of stilbenes in DMSO at a concentration of 10 mM. Compounds **1**–**6** were added to the wells of the plates in a volume of 20 μL diluted in PBS at the following final concentrations: 0.1, 1.0, and 10.0 µM (final concentration of DMSO <1%). 

Neuro-2a cells (1 × 10^4^ cells/well) were kept in a CO_2_ incubator at 37 °C for 24 h until they formed an adherent monolayer. Then, 20 μL of stilbenes solution was added to the cells. After incubation for 24 h, the medium containing the stilbenes was replaced by 100 μL of fresh DMEM medium. Then, we added 10 μL of MTT (3-(4,5-dimethylthiazol-2-yl)-2,5-diphenyltetrazolium bromide) (Sigma-Aldrich, St. Louis, MO, USA) stock solution (5 mg/mL) to each well and incubated the microplate for 4 h. After that, we added 100 μL of SDS–HCl solution (1 g SDS/10 mL dH_2_O/17 μL 6 N HCl) to each well. After incubation for 18 h, we measured the absorbance of the converted dye formazan on a Multiskan FC microplate photometer (Thermo Scientific, Waltham, MA, USA) at a wavelength of 570 nm [42]. We performed all experiments in triplicate. We expressed the cytotoxicity of the stilbenes as percentages of cell viability. 

### 4.10. In Vitro Model of PQ-Induced Neurotoxicity 

After 24 h of adhesion, we treated Neuro-2a cells (1 × 10^4^ cells/well) with stilbenes (0.01–10 μM) for 1 h and added 1 mM of PQ (Sigma-Aldrich, St. Louis, MO, USA). Cells incubated without PQ or with PQ were used as positive and negative control, respectively. After 24 h, we measured the cell viability using MTT assay. The results are presented as percentages of the positive control value. 

### 4.11. Reactive Oxygen Species (ROS) Analysis in PQ-Treated Cells

After 24 h of adhesion, Neuro-2a cells (1 × 10^4^ cells/well) were incubated with stilbenes (0.01–10 µM) for 1 h. After that, we added PQ (1 mM) to each well and incubated the cells for 3 h. To study ROS formation, we added 20 µL of 2,7-dichlorodihydrofluorescein diacetate solution (10 µM, H2DCF-DA, Molecular Probes, Eugene, OR, USA) to each well, so that the concentration was 10 µM, and kept the microplate in a CO_2_–incubator for additional 30 min at 37 °C. 

### 4.12. Mitochondrial Membrane Potential (MMP) Evaluation

We incubated the cells for 1 h in a 96-well plate (1 × 10^4^ cells/well) with stilbenes (1 and 10 µM). Then, we added PQ (500 µM) and kept the cell suspension in a CO_2_—incubator for 1 h. Cells incubated without PQ and stilbenes and with PQ only were used as positive and negative control, respectively. We added the tetramethylrhodamine methyl (TMRM) (Sigma-Aldrich, St. Louis, MO, USA) solution (500 nM) to each well and kept the cells for 30 min in a CO_2_–incubator at 37 °C. The intensity of fluorescence was measured with a PHERAstar FSplate reader (BMG Labtech, Ortenberg, Germany) at λ_ex_ = 540 nm and λ_em_ = 590 nm. We processed the data by MARS Data Analysis v. 3.01R2 (BMG Labtech, Ortenberg, Germany) and presented the results as percentages of the positive control value.

### 4.13. HSV-1 Virus and Vero Cell Culture

The HSV-1 strain L2 and Vero cell culture (kidney epithelial cells of the African green monkey Chlorocebus sp.) were obtained from the N.F. Gamaleya Federal Research Centre for Epidemiology and Microbiology (Moscow, Russia). HSV-1 was grown in Vero cells, using Dulbecco′s Modified Eagle′s Medium (DMEM, Biolot, St. Petersburg, Russia) supplemented with 10% fetal bovine serum (FBS, Biolot, St. Petersburg, Russia) and 100 U/mL of gentamycin (Dalkhimpharm, Khabarovsk, Russia), at 37 °C in a CO_2_ incubator. In the maintenance medium, the FBS concentration was decreased to 1%.

The tested compounds were dissolved in DMSO (Sigma, St. Louis, MO, USA) at a concentration 10 mg/mL and stored at −20 °C. For cytotoxicity and anti-HSV-1 activity determination, the stock solutions were diluted with DMEM so that the final concentration of DMSO was 0.5%.

### 4.14. Cytotoxicity of the Tested Compounds against Vero Cells

The cytotoxicity evaluation of the studied compounds was performed using the MTT assay, as described previously [39]. In brief, confluent Vero cells in 96-well microplates (1 × 10^4^ cells/well) were incubated with tested compounds at various concentrations (1–500 μg/mL) at 37 °C for 48 h (5% CO_2_). Untreated cells were used as controls. Then, MTT solution (methylthiazolyltetrazolium bromide, Sigma, St. Louis, MO, USA) was added to cells at a concentration of 5 mg/mL and the cells were incubated at 37 °C for 2 h. After dissolution of formazan crystals, optical densities were read at 540 nm (Labsystems Multiskan RC, Vantaa, Finland). Cytotoxicity was expressed as the 50% cytotoxic concentration (CC_50_) of the tested compound that reduced the viability of treated cells by 50% compared with control cells. [42]. Experiments were performed in triplicate and repeated three times.

### 4.15. Anti-HSV-1 Activity of Stilbenes 

The anti-HSV-1 activity of stilbenes was evaluated using cytopathic effect (CPE) inhibition assay in Vero cells. We infected the monolayer of cells grown in 96-well plates (1 × 10^4^ cells/well) with 100 µL/well of virus suspension (100 TCD_50_/_mL_) and we simultaneously treated it with stilbenes **1**–**6** (100 µL/well) at various concentrations (from 1 to 400 μg/mL) for one hour at 37 ◦C. After virus absorption, we removed the mixture of virus and stilbenes, washed the cells, and added the maintenance medium with 1% FBS. We kept the plates at 37 °C in a CO_2_-incubator for 48 h until 90% CPE was observed in virus control compared to cell control. We used MTT assay to evaluate the antiviral activity of stilbenes. The viral inhibition rate (IR, %) was calculated according to Equation (3) [43]:(3)$$\mathrm{IR},\;\%=\frac{A_{tv}-A_{cv}}{A_{cd}-A_{cv}}\times 100,$$where:*A_tv_* is the absorbance of cells infected with virus and treated with a polyphenolic compound;*A_cv_* is the absorbance of the untreated virus-infected cells;*A_cd_* is the absorbance of control (untreated and non-infected) cells.

The concentration of the compound that reduced the virus-induced CPE by 50% (50% inhibitory concentration, IC_50_) was calculated using a regression analysis of the dose–response curve [44]. The selectivity index (SI) was calculated as the ratio of CC_50_ to IC_50_. Experiments were repeated three times.

### 4.16. Extraction of HSV-1 DNA from Infected Vero Cells

For the viral DNA extraction assay, Vero cells grown in 96-well plates were infected with HSV-1 and simultaneously treated with polyphenolic compounds and incubated at 37 °C in a CO_2_-incubator for 48 h. After incubation, the culture media and the cells scraped from the plate were transferred to centrifuge tubes. The cell debris was removed by centrifugation at 300× *g* for 10 min. Then, supernatant was collected and kept at −20 °C. The HSV-1 DNA was extracted from supernatant by using the AmpliSens^®^ DNA-sorb-AM (K1-12-100-CE) nucleic acid extraction kit (Moscow, Russia) according to the manufacturer′s instructions. The supernatants were treated with a lysis solution that contained chaotropic agent (guanidine chloride) in the presence of sorbent (silica particles). As the elution solution was added, the DNA was adsorbed on silica particles and then, separated from the sorbent particles by centrifugation.

### 4.17. DNA HSV Detection by the Real-Time Polymerase Chain Reaction (PCR) Method

The DNA HSV was detected in the obtained DNA samples using the AmpliSens^®^ HSV I, II-FRT-100F PCR kit (Moscow, Russia) on a real-time PCR instrument Rotor-Gene Q (Qiagen, Hilden, Nordrhein-Westfalen, Germany) according to the manufacturer′s instructions. The HSV-I detection by this PCR kit was based on the amplification of the pathogen genome specific region (DNA-target and target gene–gB gene) using specific HSV I primer. In the real-time PCR, the amplified product is detected with the use of fluorescent dyes. The PCR reaction was carried out under the following conditions: initial denaturation at 95 °C for 15 min; then 5 cycles: at 95 °C for 5 s, at 60 °C for 20 s, at 72 °C for 15 s; then 40 cycles: at 95 °C for 5 s, at 60 °C for 20 s, at 72 °C for 15 s. A negative sample was used as the amplification control for each run. The threshold cycle number, Ct, was measured as the PCR cycle, where the amount of amplified target reached the threshold value. The reduction in the HSV-1 DNA levels in culture supernatants was assessed by the change in the threshold cycle (Δ*C_t_*) (Equation (4)):(4)$$\Delta C_{t}=C_{ttv}-C_{tcv}$$where:*C_ttv_* is the average *C_t_* value for the infected samples after treatment with polyphenolic compounds;*C_tcv_* corresponds to the average *C_t_* value for the virus control.

For each compound concentration, the viral inhibition rate (IR, %) was calculated according to Equation (5): (5)$$\mathrm{IR},\;\%=\frac{C_{ttv}-C_{tcv}}{C_{tcc}-C_{tcv}}\times 100,$$where:*C_ttv_* is the average *C_t_* value for the infected samples after treatment with polyphenolic compound;*C_tcv_* corresponds to the average *C_t_* value for the virus control;*C_tcc_* corresponds to the average *C_t_* value for the cell control.

The concentration of the compound that reduced the level of HSV-1 DNA by 50% (IC_50_) and the SI of the compound were calculated.

### 4.18. Statistical Analysis

All the experiments were carried out in triplicate. Student’s *t*-test was performed using SigmaPlot 14.0 (Systat Software Inc., San Jose, CA, USA) to determine statistical significance.

## 5. Conclusions

We isolated a new isoflavanostilbene 3*R*4*S*-maackiapicevstitol (**1**) as a mixture of two stable conformers **1a** and **1b** as well as five previously known dimeric and monomeric stilbens: piceatannol (**2**), maackin (**3**), scirpusin A (**4**), maackiasine (**5**), maackolin (**6**) from *M. amurensis* heartwood.

We showed that Maksar^®^ and its components possessed significant antioxidant properties and reduced the level of intracellular ROS in infected cells, which resulted in the inhibition of HSV-1 and reduced neurotoxicity in a model of Parkinson′s disease. Thus, Maksar^®^ and its components that possessed significant neuroprotective potential and moderate antiherpetic properties were reported in this study. These results open perspectives to investigate the potential of Maksar^®^ for new medical applications.

## Figures and Tables

**Figure 1 molecules-28-02593-f001:**
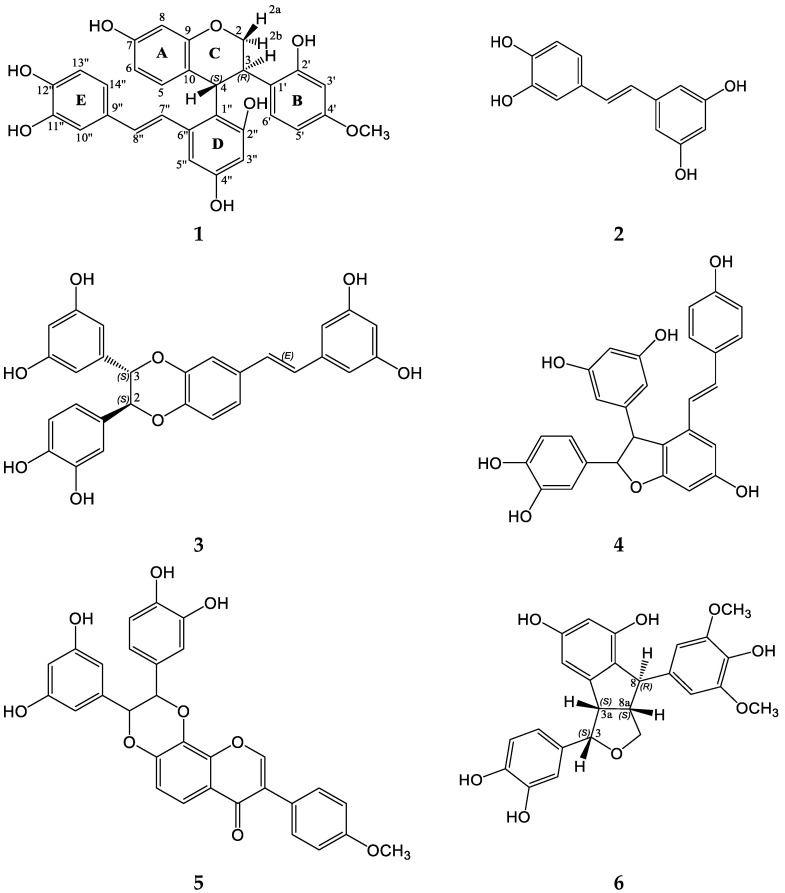
Chemical structures of polyphenolic compounds isolated from *M. amurensis* heartwood.

**Figure 2 molecules-28-02593-f002:**
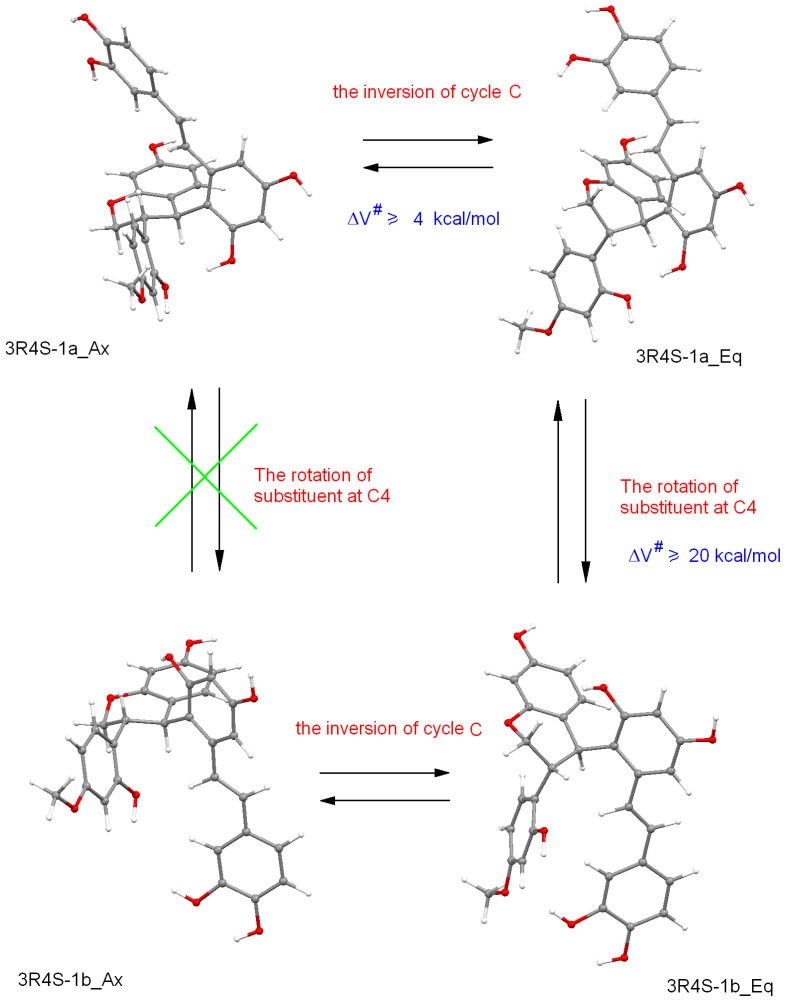
Two main intramolecular rearrangements of 3*R*,4*S* stereoisomer of **1**.

**Figure 3 molecules-28-02593-f003:**
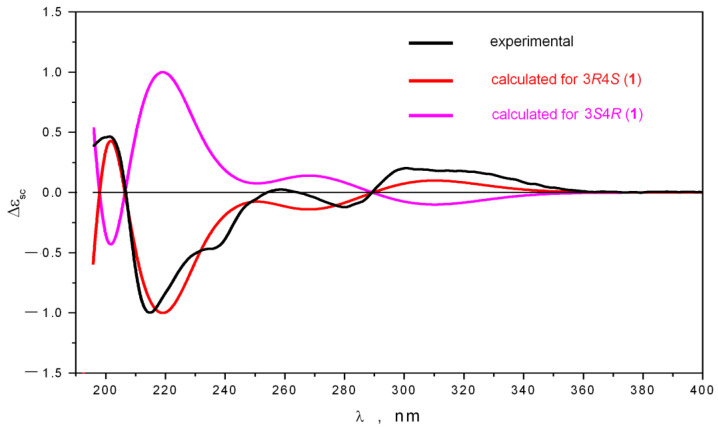
The comparison of the experimental ECD spectrum of **1** with the spectra calculated for the 3*R*4*S* and 3*S*4*R* stereoisomers of **1**.

**Figure 4 molecules-28-02593-f004:**
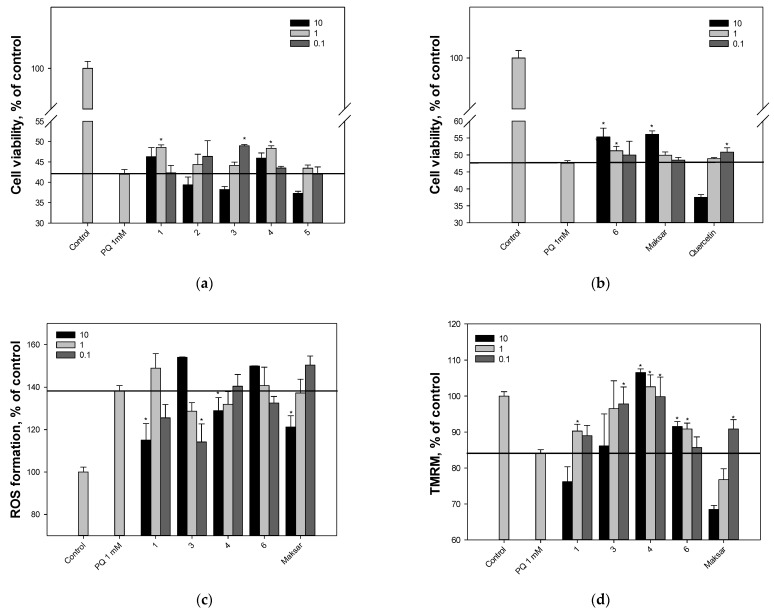
The effect of stilbenes from *M. amurensis* on cell viability (**a**,**b**), intracellular ROS levels (**c**), and mitochondrial membrane potential (**d**) in PQ-treated Neuro-2a cells (1 mM). The cell viability after treatment with stilbenes and PQ was measured using the MTT assay. Stilbenes **1**–**6** were used at concentrations 10, 1, 0.1 μM. Maksar^®^ was used at concentrations 10, 1, 0.1 μg/mL. Data are presented as means ± SEM of three independent replicates. (*) indicated significant difference versus PQ-treated cells (*p* < 0.05).

**Figure 5 molecules-28-02593-f005:**
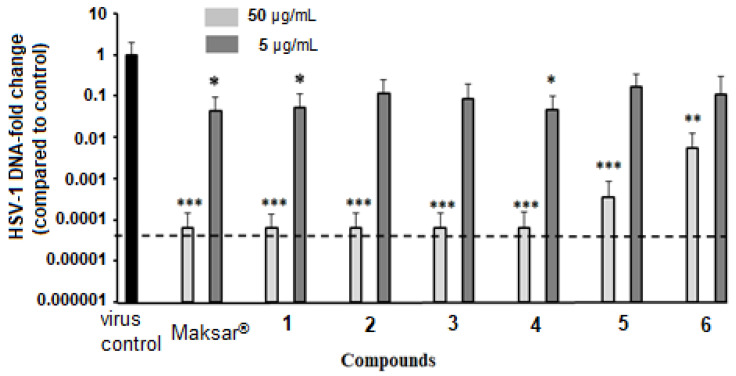
Anti-HSV-1 activity of polyphenolic compounds from *M. amurensis* (RT-PCR). The tested compounds at concentrations of 50 μg/mL and 5 μg/mL were added to Vero cells simultaneously with the virus. The relative changes in HSV-1 DNA levels were calculated using the 2^−ΔCt^ method and reported as the fold reduction relative to virus control (control = 1). The dashed line represents DMSO-treated cells (cell control). Data are expressed as means ± SD of three independent experiments. (*), (**), (***) indicated significant difference versus PQ-treated cells (*p* < 0.05, *p* < 0.01, *p* < 0.001).

**Table 1 molecules-28-02593-t001:** ^1^H (700 MHz), ^13^C (175 MHz), HMBC, COSY, and ROESY NMR data for compound **1a** (*δ* in ppm, *J* in Hz, acetone-*d*_6_).

Position	^13^C	^1^H	HMBC	COSY	ROESY
2a 2b	71.6	4.24, dd, *J* = 10.3, 3.5 1H	C-3, 4, 9, 1′	H-2b, 3	H-2b, 3
		4.03, t, *J* = 10.3, 1H	C-3, 4	H-2a, 3	H-2a, 4
3	39.0	4.16, td, *J* = 11.6, 3.5 1H	C-2, 4, 1′, 2′, 6′ (weak), 1″	H-2, 4	H-7″, 2a
4	35.8	5.36, d, *J* = 11.6, 1H	C-2 (weak), 3, 5 (weak), 9 (weak), 1″, 2″, 6″	H-3	H-2b, H-6′
5	130.1	6.64, d, *J* = 8.4, 1H	C-4, 7, 9	H-6	H-6
6	109.0	6.29 dd, *J* = 8.4, 2.4 1H	C-8, 10	H-5, 8	H-5
7	156.8				
8	103.2	6.42, d, *J* = 2.4, 1H	C-6, 7, 9, 10	H-6	
9	156.4				
10	119.5				
1′	119.0				
2′	156.8				
3′	101.8	6.30, d, *J* = 2.4, 1H	C-1′, 2′, 4′, 5′,	H-5′	OCH_3_-4′
4′	159.8				
5′	104.9	6,27, dd, *J* = 8.5, 2.4 1H	C-1′, 3′, 4′	H-3′, 6′	OCH_3_-4′
6′	129.8	7.25, d, *J* = 8.5, 1H	C-3, 2′, 4′	H-5′	H-4
1″	119.5				
2″	157.4				
3″	101.7	6.26, d, *J* = 2.5, 1H	C-1″, 2″, 4″, 5″	H-5″	
4″	156.6				
5″	106.0	6.51, d, *J* = 2.5, 1H	C-1″, 3″, 4″, 7″	H-3″	
6″	139.6				
7″	126.2	6.82, d, *J* = 16.0, 1H	C-1″, 5″, 6″ (weak), 9″	H-8″	H-3
8″	129.0	6.57, d, *J*= 16.0, 1H	C-6″, 9″, 10″, 14″	H-7″	
9″	131.2				
10″	114.0	6.80, d, *J* = 2.4, 1H	C-8″, 11″, 12″, 14″	H-14″	
11″	145.6				
12″	145.4				
13″	115.7	6.73, d, *J* = 8.1, 1H	C-9″, 11″, 12″	H-14″	
14″	119.2	6.67, dd, *J* = 8.1, 2.4 1H	C-8″, 10″, 12″	H-10″, 13″	
OCH_3_-4′	54.9	3.63, s, 3H	C-4′		H-3′, 5′

**Table 2 molecules-28-02593-t002:** ^1^H (700 MHz), ^13^C (175 MHz), HMBC, COSY, and ROESY NMR data for compound **1b** (*δ* in ppm, *J* in Hz, acetone-*d*_6_).

Position	^13^C	^1^H	HMBC	COSY	ROESY
2a 2b	70.6	4.20, dd, *J* = 10.3, 3.2 1H	C-3, 4, 9, 1′	H-2b, 3	H-2b, 3
		4.44, t, *J* = 10.3, 1H	C-4 (weak)	H-2a, 3 (weak)	H-2a
3	38.4	4.36, td, *J* = 10.9, 3.2 1H	C-2 (weak), 2′ (weak), 6′ (weak), 1″ (weak)	H-2a, 2b(weak), 4	H-2a
4	39.7	4.95, d, *J* = 10.9, 1H	C-3, 5 (weak), 1″, 2″, 6″	H-3	H-7″
5	129.3	6.50, d, *J* = 8.8, 1H	C-4, 7, 9	H-6	H-6
6	108.3	6.20 dd, *J* = 8.8, 2.4 1H	C-8, 10,	H-5, 8	H-5
7	156.6				
8	103.1	6.27, d, *J* = 2.4, 1H	C-6, 10	H-6	
9	156.3				
10	119.5				
1′	119.7				
2′	157.1				
3′	102.2	6.40, d, *J* = 2.4, 1H	C-1′, 2′, 4′, 5′	H-5′	OCH_3_-4′
4′	159.9				
5′	105.0	6,22, dd, *J* = 8.5, 2.4 1H	C-1′, 3′, 4′	H-3′, 6′	H-6′, OCH_3_-4′
6′	131.1	6.99, d, *J* = 8.5, 1H	C-3, 2′, 4′	H-5′	H-5
1″	119.4				
2″	157.1				
3″	103.6	6.20, d, *J* = 2.4, 1H	C-1″, 2″, 4″, 5″	H-5″	
4″	156.8				
5″	104.8	6.44, d, *J* = 2.4, 1H	C-1″, 3″, 4″, 7″	H-3″	
6″	141.7				
7″	126.1	7.25, d, *J* = 15.9, 1H	C-1″, 5″, 9″	H-8″	H-4
8″	131.4	6.62, d, *J* = 15.9, 1H	C-6″, 7″(weak), 9″, 10″, 14″	H-7″	H-10″
9″	131.0				
10″	113.3	7.09, d, *J* = 1.9, 1H	C-8″, 11″, 12″, 14″	H-14″	H-8″
11″	145.8				
12″	145.6				
13″	115.7	6.78, d, *J* = 8.1, 1H	C-9″, 11″, 12″	H-14″	
14″	119.9	6.84, dd, *J* = 8.5, 1.9 1H	C-8″, 10″, 12″	H-10″, 13″	
OCH_3_-4′	54.9	3.61, s, 3H	C-4′		H-3′, 5′

**Table 3 molecules-28-02593-t003:** DPPH scavenging activity and FRAP of compounds **1**–**6**.

Compound	DPPH Scavenging Effect		FRAP	
	IC_50_ µM, 20 min	IC_50_ µg, 20 min	C_Fe2+_(µM)/C_polyphenolic compound_ (µM)	C_Fe2+_(µM)/C_polyphenolic compound_ (μkg/mL)
Quercetin	9.3 ± 0.4	2.81 ± 0.12	5.53 ± 0.55	18.3 ± 1.82
Ascorbic acid	33.1 ± 2.8	5.83 ± 0.49	3.58 ± 0.29	20.34 ± 0.16
**1**	4.3 ± 0.5 *	2.25 ± 0.25	12.36 ± 1.38 **	24.04 ± 2.68 *
**2**	4.3 ± 0.5 *	1.05 ± 0.12 **	15.71 ± 1.43 ***	64.38 ± 5.86 ***
**3**	2.7 ± 0.3 **	1.31 ± 0.15 **	10.11 ± 0.93 **	20.80 ± 1.91
**4**	3.2 ± 0.4 **	1.50 ± 0.19 *	11.44 ± 1.41 **	24.34 ± 3.00 *
**5**	2.5 ± 0.3 **	1.32 ± 0.16 **	2.50 ± 0.38	4.75 ± 0.72
**6**	2.0 ± 0.3 ***	0.90 ± 0.14 ***	23.10 ± 2.65 ***	51.11 ± 5.86 ***
**Maksar^®^**	-	9.9 ± 0.94 *	-	3.5 ± 0.60*

Data are presented as the mean values ± SEM, n = 3, *** *p* < 0.001, ** *p* < 0.005, * *p* < 0.05 compared to quercetin.

**Table 4 molecules-28-02593-t004:** Cytotoxicity and anti-HSV-1 activity of polyphenolic compounds from *M. amurensis* (CPE assay).

Compounds	CC_50_		IC_50_		SI
	µg/mL	µM	µg/mL	µM	
**Maksar^®^**	1211.7 ± 133		13.9 ± 1.8		87.2 ± 11.3
**1**	162.4 ± 17.9	315.9 ± 34.7	14.0 ± 1.5	27.2 ± 3.0	11.6 ± 1.3
**2**	250.2 ± 30.0	1025.4 ± 123.0	22.0 ± 2.6	90.2 ± 10.8	11.4 ± 1.4
**3**	269.6 ± 35.0	554.7 ± 72.1	17.6 ± 2.3	36.2 ± 4.7	15.3 ± 1.9
**4**	199.4 ± 23.9	424.2 ± 50.9	13.8 ± 1.6	29.4 ± 3.5	14.4 ± 1.7
**5**	156.4 ± 18.8	297.3 ± 35.7	23.7 ± 2.8	45.0 ± 5.4	6.6 ± 0.8
**6**	256.4 ± 33.3	567.2 ± 73.7	40.2 ± 5.2	88.9 ± 11.5	6.4 ± 0.8
**Acyclovir**	>1000	>4000	2.1 ± 0.3	9.3 ± 1.3	430

Values are presented as the means ± SD of three or more independent experiments. Acyclovir^®^ was used as the positive control. SI = CC_50_/IC_50._

## Data Availability

The data are contained within the article and Appendix A.

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
