# Peer review of "Neuroprotective and Antiherpetic Properties of Polyphenolic Compounds from Maackia amurensis Heartwood"

_molecules, 2023, doi:10.3390/molecules28062593_

Round 1
Reviewer 1 Report
This study investigated the neuroprotective and antiherpetic effects of phenolic compounds isolated from Maackia amurensis heartwood. The results are well-detailed, but I think the authors should provide more details on how these compounds showed the above effects. More importantly, how these compounds are correlated to different activities and their exact mechanism of action is not detailed enough. Moreover, in the abstract, the authors should mention the techniques that were followed to isolate phenolic compounds. My other comments are:
Lines 18 and 668, is it “slilbens” or “stilbenes”?
Line 19, add “and” before maackolin
Line 21, One should define “CPE” first
Line 40, missing dot (.) after “stilbenes [1]”
Lines 43, 45, and throughout the whole manuscript, remove space between the two reference numbers.
Line 68, remove dot (.) and add a comma between 21 and 21
Line 309, why quercetin and ascorbic acid were used as reference compounds? An explanation should be provided.
Line 310, remove µM after 2.0
Author Response
We are grateful to the Reviewer for the effort in improving our manuscript. Here are our replies to the review’s comments:
|
review’s comments |
our answer |
|
1. The results are well-detailed, but I think the authors should provide more details on how these compounds are correlated to different activities and their exact mechanism of action is not detailed enough. |
For the neuroprotective activity, we studied two mechanisms: ROS formation and the effect of polyphenolic compounds on mitochondrial membrane potential. We showed that these mechanisms can be involved in the neuroprotective activity of the studied polyphenolic compounds. These data are presented in the manuscript. For the antiviral activity, we are planning to carry out a number of experiments to study the effect of polyphenolic compounds from M. amurensis heartwood on different stages of viral replication in Vero cells. Besides, we will increase the number of polyphenolic compounds under study in order to make conclusion on the structure-activity relationship. The results will be published in our future papers. |
|
2. Moreover, in the abstract, the authors should mention the techniques that were followed to isolate phenolic compounds. |
We have made the necessary corrections in Abstract. |
|
3. My other comments are: Lines 18 and 668, is it “slilbens” or “stilbenes”? (1) Line 19, add “and” before maackolin (2) Line 21, One should define “CPE” first (3) Line 40, missing dot (.) after “stilbenes [1]” (4) Lines 43, 45, and throughout the whole manuscript, remove space between the two reference numbers. (5) Line 68, remove dot (.) and add a comma between 21 and 21 (6) Line 309, why quercetin and ascorbic acid were used as reference compounds? An explanation should be provided. (7) Line 310, remove µM after 2.0 (8)
|
We have made the necessary corrections (1-8) throughout the text of the manuscript.
Quercetin and ascorbic acid arte often used as reference compounds to evaluate DPPH scavenging effect and FRAP of various natural polyphenolic compounds [1. Katrin Sak. Dependence of DPPH Radical Scavenging Activity of Dietary Flavonoid Quercetin on Reaction Environment // Mini Rev Med Chem. Vol. 14 (60), 2014; P.: 494 – 504. DOI: 10.2174/1389557514666140622204037; 2. Sridevi Chigurupati , Sohrab Akhtar Shaikh , Jahidul Islam Mohammad, Kesavanarayanan Krishnan Selvarajan , Appala Raju Nemala, Chu How Khaw, Chun Foo Teoh , Ting Hei Kee. In vitro antioxidant and in vivo antidepressant activity of green synthesized azomethine derivatives of cinnamaldehyde // Indian J Pharmacol. Vol. 49 (3), 2017; P.: 229 – 235 DOI: 10.4103/ijp.IJP_293_16 ]
|

Reviewer 2 Report
In the present study titeled « Neuroprotective and antiherpetic properties of polyphenolic 2 compounds from Maackia amurensis heartwood » , the authors disclosed the isolation of six polyphenolic compounds from an ethanol-chloroform extract of Maackia amurensis heartwood, among which one compound is new, not previously described in the literature. A number of experiments were carried out to study biological activity of isolated compounds - antioxidant, cytotoxic, antiherpetic and neuroprotective. The research material presented in the article is carried out at a high experimental level, the results are beyond doubt. I would like to separately note the excellent work on establishing the structure of compound 1, using various approaches.
Main concerns
1. Title. Why are only two properties of isolated polyphenolic compounds included in the Title? In addition, the Title does not reflect the part of the work devoted to the establishment of the structure of compound 1.
2. In my opinion, it would be informative to give the HPLC spectra of the total extract and fraction 12 and 14, from which the polyphenolic compounds were isolated.
Minor concerns
1. Please check throughout the text: sTilbene (lines 18, 80, 668)
2. Figure 1 and line 85: “1a, 1b” should be replaced by “1 (compound 1)” or “1 (mixture of conformers 1a and 1b)”
3. Part 4.2 should be placed after Part 4.5 because Parts 4.3 to 4.5 describe the methods applicable in Part 4.2.
Author Response
We are grateful to the Reviewer for the effort in improving our manuscript. Here are our replies to the review’s comments:
|
Reviewer 2 |
|
|
1. Title. Why are only two properties of isolated polyphenolic compounds included in the Title? In addition, the Title does not reflect the part of the work devoted to the establishment of the structure of compound 1. |
We suppose that the title of the manuscript should not be changed, because: 1. The main purpose of our work was to study the neuroprotective and antiherpetic activities of polyphenolic compounds from Maackia amurensis heartwood. The neuroprotective properties of polyphenolics are mainly due to their antioxidant activity. 2. It is not necessary to include the structure establishment of 1 into the title, because it is obvious that the structures of polyphenolics must be established before their biotesting. |
|
2. In my opinion, it would be informative to give the HPLC spectra of the total extract and fraction 12 and 14, from which the polyphenolic compounds were isolated.
|
We gave the HPLC profiles of fractions 12 and 14 in Suppplementary data (S87, S88), from which the polyphenolic compounds were isolated. The HPLC profile of the total extract from M. amurensis was published previously [Fedoreev, S.A.; Kulesh, N.I.; Glebko, L.I.; Pokushalova, T.V.; Veselova, M.V.; Saratikov, A.S.; Vengerovskii, A.I.; Chuchalin, V.S. Maksar: a preparation based on Amur Maackia. Pharm. Chem. J. 2004, 38, 605–610. DOI: 10.1007/s11094-005-0039-6]. |
|
Minor concerns 1. Please check throughout the text: sTilbene (lines 18, 80, 668) 2. Figure 1 and line 85: “1a, 1b” should be replaced by “1 (compound 1)” or “1 (mixture of conformers 1a and 1b)” |
We made the necessary corrections (1 and 2) throughout the text. |

Reviewer 3 Report
Darya Tarbeeva and her co-authors have written a very interesting manuscript on the isolation and structure elucidation of polypenolic compounds with interesting bioactivities.
The article is written in a understandable English and fulfills all the requirements of a good scientific work.
In the appendix, the 1D and 2D spectra of 1a/1b are shown in great detail. This is very commendable! I would have reduced the integral curve in the proton spectrum so that the entire curve is visible and does not protrude out of the drawing area.
I would also have labelled all peaks in the 13C NMR spectrum (e.g. peaks at about 119 ppm).
Even as the spectra are presented in such detail, it is very difficult to check them. Also because it is unfortunately a mixture of 1a and 1b.
Most of the assignments seem plausible and comprehensible. There is one small thing the authors should please check: The signal at 4.16 ppm for proton H3 (cmp. 1a), described as dd (11.6 / 3.5) looks in the 1H and in the HSQC more like a td with a large coupling constant (ca 11 Hz) and a small one (ca 2 Hz). The same is true for signal at 4.36 ppm (H3 cmp. 1b). I am also a little bit surprised that the 13C signal at 38.4 ppm for C3 (cmp. 1A) is not visible in the 13C NMR spectrum.
Just a hint for future work: I would have recorded an additional HSQC with a smaller spectral width (sw2 40 ppm, o2p 120 ppm, td2 256) just to have a much better resolution in f1. The same I would have done with the HMBC.
Have the authors ever tried to record NMR spectra at higher temperature?
It would be a good idea to have the English improved by a native speaker. The main problem besides wrong clause position is the lack of definite and indefinite articles. I mean, article is easy to read and understand, but its not wrong to improve it.
just some expamples from the first 3 pages
line 18: stilbens
line 36: is used to produce the drug
line 36: was developed at the G.B. Elyakov Pacific Institute of Bioorganic Chemistry
line 40: dimeric stilbenes [1]. In contrast
line 56: and the isoflavones calicosin and 8-O-methylretusin as well as the pterocarpan
line 57: The isoflavones genistein
line 72: properties of the polyphenols from
With a clear conscience I can propose the manuscript for publication after small revisions.
Author Response
We are grateful to the Reviewer for the effort in improving our manuscript. Here are our replies to the review’s comments:
|
Reviewer 3 |
|
|
I would have reduced the integral curve in the proton spectrum so that the entire curve is visible and does not protrude out of the drawing area.
|
We have corrected the 1H NMR spectra of 1 in Supplementary data according to the review’s comments. |
|
I would also have labelled all peaks in the 13C NMR spectrum (e.g. peaks at about 119 ppm).
|
It is difficult to label all peaks in the area at 119.5, because their chemical shift values are very close to each other. However, it did not affect the structure elucidation process of compound 1. |
|
Even as the spectra are presented in such detail, it is very difficult to check them. Also because it is unfortunately a mixture of 1a and 1b. Most of the assignments seem plausible and comprehensible. There is one small thing the authors should please check: The signal at 4.16 ppm for proton H3 (cmp. 1a), described as dd (11.6/3.5) looks in the 1H and in the HSQC more like a td with a large coupling constant (ca 11 Hz) and a small one (ca 2 Hz). The same is true for signal at 4.36 ppm (H3 cmp. 1b). I am also a little bit surprised that the 13C signal at 38.4 ppm for C3 (cmp. 1A) is not visible in the 13C NMR spectrum. Just a hint for future work: I would have recorded an additional HSQC with a smaller spectral width (sw2 40 ppm, o2p 120 ppm, td2 256) just to have a much better resolution in f1. The same I would have done with the HMBC. |
We thank the reviewer for helping us correct the mistake in describing the signals in the NMR spectra. Although the spectrum is rather complicated, we managed to establish the structure of compound 1. We agree that the signal at 4.16 ppm for proton H3 (cmp. 1a) looks more like td. We made the necessary corrections in Table 1 and throughout the text of the manuscript.
Althogh the signal at 38.4 ppm for C-3 (cmp. 1A) is not visible in the 13C NMR spectrum, we observed it in the HMBC spectrum of compound 1. The reviewer’s comments will be useful for us in our future work. |
|
Have the authors ever tried to record NMR spectra at higher temperature?
|
In our opinion, the registration of NMR and ECD spectra for 1 at different temperatures can not significantly clarify the details of 1а ↔ 1b transformation. Such experiments are very informative in the cases, when the compound under study is relatively rigid and there are only one or two large-amplitude motions degrees of freedom (LAM), the movement along which may be treated separately from the movements along other degrees of freedom. Compound 1 belongs to the class of structurally very flexible compounds, for which a big number of LAM motions may proceed at one and the same time (see Figure S71-83, Supplementary data). The IR spectrum calculated theoretically for 1 contains more then 30 vibrational modes with frequencies n £ 350 cm-1. Fifteen of them have frequencies n £ 150 cm-1, and for six modes n £ 50 cm-1. The heating of the sample will lead to the effective growth of populations of the excited states of these modes and, as a result, - to the growth of the so-called “vibrational contribution” to the values of spectral parameters. To extract the contribution of the 1а ↔ 1b transformation from the total experimentally detected temperature dependence one have to use complicated theoretical models, in which the movements along all LAM degrees of freedom are treated simultaneously (with the account of anharmonicity of corresponding potentials). At present such modeling is far and far away from the routine and may be seen as a task for separate investigation. We recalculate the height of potential barrier for 1а ↔ 1b transformation with 6-311++G(d,p) basis set: DV#(B3LYP/6-311++G(d,p)_PCM) » 21 kcal/mol.
This confirms our previous estimation DV# ³ 20 kcal/mol. The lifetimes of 1a and 1b conformers are t » 3 c for T = 330 K. This is why we will observe 1a and 1b as two separate compounds in the NMR spectra. Besides, the boiling point of acetone is 56°C (329 K), so we have to record NMR spectra at temperatures lower than 330 K. |
|
It would be a good idea to have the English improved by a native speaker. The main problem besides wrong clause position is the lack of definite and indefinite articles. I mean, article is easy to read and understand, but its not wrong to improve it. just some expamples from the first 3 pages line 18: stilbens line 36: is used to produce the drug line 36: was developed at the G.B. Elyakov Pacific Institute of Bioorganic Chemistry line 40: dimeric stilbenes [1]. In contrast line 56: and the isoflavones calicosin and 8-O-methylretusin as well as the pterocarpan line 57: The isoflavones genistein line 72: properties of the polyphenols from With a clear conscience I can propose the manuscript for publication after small revisions. |
We made the necessary corrections throughout the text of the manuscript. |

Round 2
Reviewer 1 Report
The authors have now revised the manuscript accordingly, and it is ready to proceed to the next step.
Reviewer 2 Report
Please correct typos:
line 20: "silcagel"
line 104: 'correl;ation'